developmental biology/biophysics/computational biology

differential growth, interkinetic nuclear migration, live imaging, mathematical model, spiral shape formation, tissue morphogenesis

**Author for correspondence:**
Tsuyoshi Hirashima
e-mail: hirashima.tsuyoshi.2 m@kyoto-u.ac.jp

# Stalling interkinetic nuclear migration in curved pseudostratified epithelium of developing cochlea

Mamoru Ishii[1], Tomoko Tateya[2,5], Michiyuki Matsuda[1,3,4] and Tsuyoshi Hirashima[1,6,7]

[1]Graduate School of Biostudies, [2]Department of Otolaryngology-Head and Neck Surgery, Graduate School of Medicine, [3]Department of Pathology and Biology of Diseases, Graduate School of Medicine, and [4]Institute for Integrated Cell-Material Sciences, Kyoto University, Kyoto, Japan
[5]Department of Speech and Hearing Sciences and Disorders, Faculty of Health and Medical Sciences, Kyoto University of Advanced Science, Kyoto, Japan
[6]The Hakubi Center, Kyoto University, Yoshida-Konoe-cho, Sakyo-ku, Kyoto 606-8501, Japan
[7]Japan Science and Technology Agency, PRESTO, Kawaguchi, Japan

MI, 0000-0002-3399-502X; TT, 0000-0002-0342-3688;
MM, 0000-0002-5876-9969; TH, 0000-0001-7323-9627

The bending of epithelial tubes is a fundamental process in organ morphogenesis, driven by various multicellular behaviours. The cochlea in the mammalian inner ear is a representative example of spiral tissue architecture where the continuous bending of the duct is a fundamental component of its morphogenetic process. Although the cochlear duct morphogenesis has been studied by genetic approaches extensively, it is still unclear how the cochlear duct morphology is physically formed. Here, we report that nuclear behaviour changes are associated with the curvature of the pseudostratified epithelium during murine cochlear development. Two-photon live-cell imaging reveals that the nuclei shuttle between the luminal and basal edges of the cell is in phase with cell-cycle progression, known as interkinetic nuclear migration, in the flat region of the pseudostratified epithelium. However, the nuclei become stationary on the luminal side following mitosis in the curved region. Mathematical modelling together with perturbation experiments shows that this nuclear stalling facilitates luminal-basal differential growth within the epithelium, suggesting that the nuclear stalling would contribute to the bending of the pseudostratified epithelium during the cochlear duct development. The findings suggest a possible scenario of differential growth which sculpts the tissue shape, driven by collective nuclear dynamics.

# 1. Introduction

Tube bending is a fundamental motif in tissue morphogenetic processes and is often observed in developing organs and organisms [1–6]. The bending process, which can be described by the nature of asymmetric deformation at tissue scale, is facilitated by the differential growth of an epithelial tube along with an axis perpendicular to the longitudinal axis, as shown in various tissues [4,7,8]. For example, asymmetric profiles of cellular events, such as cell elongation and proliferation, between the opposite sides across the central axis of the tube generate differences in local tissue growth, with the growth rate on one side being faster than that on the other side. The imbalance of local tissue growth, regardless of whether active or passive, contributes to the elastic bending of developing tubes and may originate in different cellular processes depending on the biological tissue being studied. Thus, clarifying the specific mechanisms underlying differential tissue growth is critical to understanding the morphogenetic processes in organ development.

The mammalian cochlear duct is a typical example of these tissues and relies on tube bending during the development of its spiral architecture. During mouse development, the epithelial cochlear duct initiates from a subregion of the otocyst and extends to form a half turn by embryonic day (E) 12, according to the tissue axes determined in an earlier phase [9,10]. The cochlear duct will then continue to bend and coil while undergoing elongation until it achieves its mature shape (figure 1a). The murine cochlear duct is tonotopically organized as with other mammals; the base of the duct is more sensitive to high-frequency sounds while the apex region is more sensitive to low-frequency sounds [11]. Previous genetic studies have uncovered the molecular basis of murine cochlear duct morphogenesis, and gene knockout studies have clarified that the elongation of the cochlear duct requires essential signalling pathways, such as sonic hedgehog signalling [12–14], fibroblast growth factor signalling [15–18] and non-canonical Wnt–planar cell polarity signalling [19–23]. Although these knockout studies have contributed to our understanding of the complex signalling environment needed to develop the cochlear duct, there are no reports on a phenotype in mutants, in which the duct bending is severely impaired without a change in the overall length of these ducts compared to the wild-type. Thus, the genetic approach has some limitations in elucidating the underlying cellular processes allowing for the asymmetric bending of the cochlear duct.

Accumulating evidence produced using live-cell imaging and genetic analysis has shown that the mediolateral intercalating active migration of cochlear duct cells occurs and contributes to longitudinal duct extension through convergent extension from E14.5 [24–27]. The convergent extension of these cellular movements drives cochlear duct elongation, but it is still uncertain whether it can contribute to duct bending. Such tissue deformation usually relies on various asymmetric properties at the cell and tissue levels, such as chirality in the subcellular localization of the cytoskeleton and/or cell shape [28,29] or heterogeneous distribution of cellular intercalation along with the mediolateral axis. Without this asymmetry cellular intercalation is simply insufficient to explain any duct bending before E14.5, as no clear mediolateral intercalation was observed between E12.5 and E14.5 [30].

We recently showed that there were more proliferating cells on the medial side of the cochlear duct apex than on the lateral side at E12.5 [30]. If we assume that cell proliferation is the main driver of local tissue growth, the higher volumetric growth observed on the medial side of these tissues would contribute to the outward bending of the duct, which contradicts physiological cochlear morphogenesis. One way to resolve this mismatch is to supply cells to the growing apex region from the base region, where there is extensive cellular proliferation. We found that multicellular oscillation flows from the base region on the lateral side of the developing cochlear duct and is driven by retrograde waves of helical activation of extracellular signal-regulated kinase (ERK) derived from the interplay between ERK activation and active cell migration [30–32]. This mechanism explains the advection of the cells on the lateral side. However, it remains unclear what cellular processes underpin the differences in mediolateral tissue growth necessary for cochlear duct bending.

Here, we find characteristic nuclear behaviours on the medial side of the cochlear duct using two-photon live imaging and demonstrate that the nuclei stall at the luminal side of the cochlear duct during up-and-down motion within the pseudostratified epithelium. We combine this data with mathematical modelling and drug perturbation and propose that nuclear stalling causes local tissue extension, suggesting that differential growth sculpts the tissue shape and is driven by collective nuclear dynamics.

# 2. Results

## 2.1. Medial epithelial layer primarily causes cochlear duct bending

The spiral form of the cochlear duct is characterized by mediolateral asymmetry (figure 1a, reproduced from [30]). Therefore, we first examined the force balance between the medial and lateral tissues by surgically

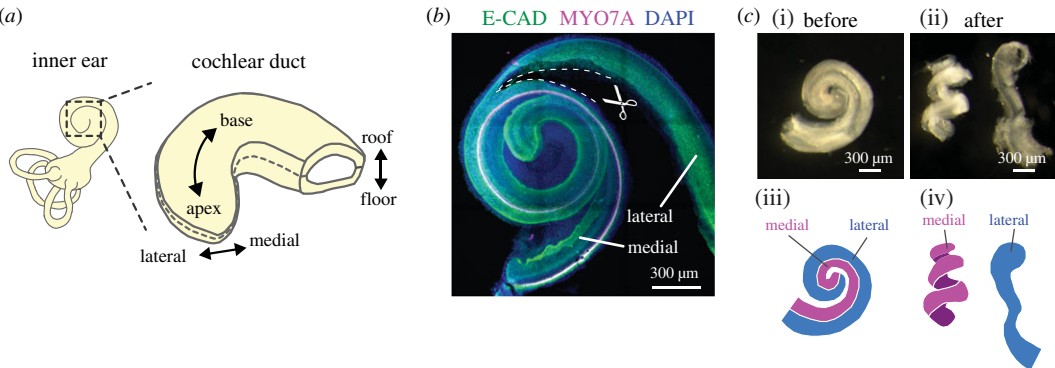

**Figure 1.** Surgical separation reveals that the bending force resides in the medial side of the developing cochlear duct. (*a*) Schematic diagrams showing the tissue axis and labels of the cochlear duct. (*b*) Immunofluorescence image of anti-E-cadherin (green) and anti-Myosin VIIA, a marker of the sensory hair cells (magenta), with nuclear counterstaining using DAPI (blue) of the cochlea at E17.5 during tissue separation. Surgical cuts are represented by dashed lines. Scale bar = 300 µm. (*c*) Images of the cochlear duct before (i, iii) and after (ii, iv) tissue separation. Stereomicroscope images (i, ii) and the corresponding illustrations of the medial (magenta, iii, iv) and lateral (blue, iii, iv) layers. N = 3 was confirmed. Scale bar = 300 µm.

separating the cochlear duct along with the lateral side of the hair cells using E17.5 cochlear tissues (figure 1*b*). The medial side includes the Kölliker's organ and the prosensory domain, while the lateral side includes the outer sulcus following separation. This manipulation led to the relaxation of the mechanical stress between the medial and lateral tissues and enabled us to inspect the internal stresses of the local tissues. The medial tissue curled immediately after separation maintaining a similar degree of curling to that described for the intact cochlear duct. However, the curvature of the lateral tissues decreased dramatically (figure 1*c*; electronic supplementary material, movie S1), indicating that the bending force is primarily generated by the medial side of these ducts at this stage of development. Moreover, even when the separated medial side was manually straightened, the tissue reverted to a curled structure within a few seconds (electronic supplementary material, movie S1). This result motivated us to explore the physical properties of the medial side of the cochlear duct during the earlier stages of development.

## 2.2. The curved medial epithelium is a proliferative pseudostratified tissue

Next, we went on to examine the morphology of the developing cochlear duct between E12.5 to E15.5, using organ-scale three-dimensional imaging. We used horizontal sections to project images along the roof–floor axis [30] and quantified the curvature and thickness of the epithelial layers on both the medial and lateral sides. Hereafter, we refer to the epithelial layers on the medial and lateral sides as the medial and lateral epithelial layers (MEL and LEL), respectively (figure 2*a*). We found that the curvature and thickness of the MEL changed remarkably more than 100 µm away from the apex tip of the duct along with the MEL at any time stage, while the curvature and thickness of the LEL remained constant with small variations (figure 2*b*, graphs describing the LEL from E12.5 to E14.5 were adopted from our previous study [30] for the comparison between LEL and MEL). These observations prompted us to explore the structure and dynamics of the MEL at single-cell resolution.

Zooming in on the epithelial layer clarified that most of the nuclei in the LEL align within a few nuclei diameters of each other (approx. 20 µm) (figure 2*c*i). However, nuclei in the MEL are distributed broadly across an approximately 50 µm range, and mitotic cell rounding occurs on the luminal side of the MEL (figure 2*c*ii). We also found that any nuclei stained positive for the M phase marker, phospho-histone H3 (pHH3), only localized on the luminal side of the epithelial layers. By contrast, nuclei in the S phase cells, labelled with a short pulse of ethynyl deoxyuridine (EdU), were predominantly distributed on the basal side of the cells regardless of whether they were in the LEL or MEL (figure 2*d*). In addition, mosaic cell labelling revealed that single-cell bridges extend between the luminal and basal edges of the MEL via long protrusions (figure 2*e*), clearly indicating that the MEL is a proliferative pseudostratified epithelial tissue.

## 2.3. Luminal nuclear stalling occurs in the curved pseudostratified epithelium

We then focused on the cellular dynamics of the MEL using a two-photon microscope. To allow for the live imaging of the MEL, the outer cartilaginous capsule shell that will eventually develop into the bony

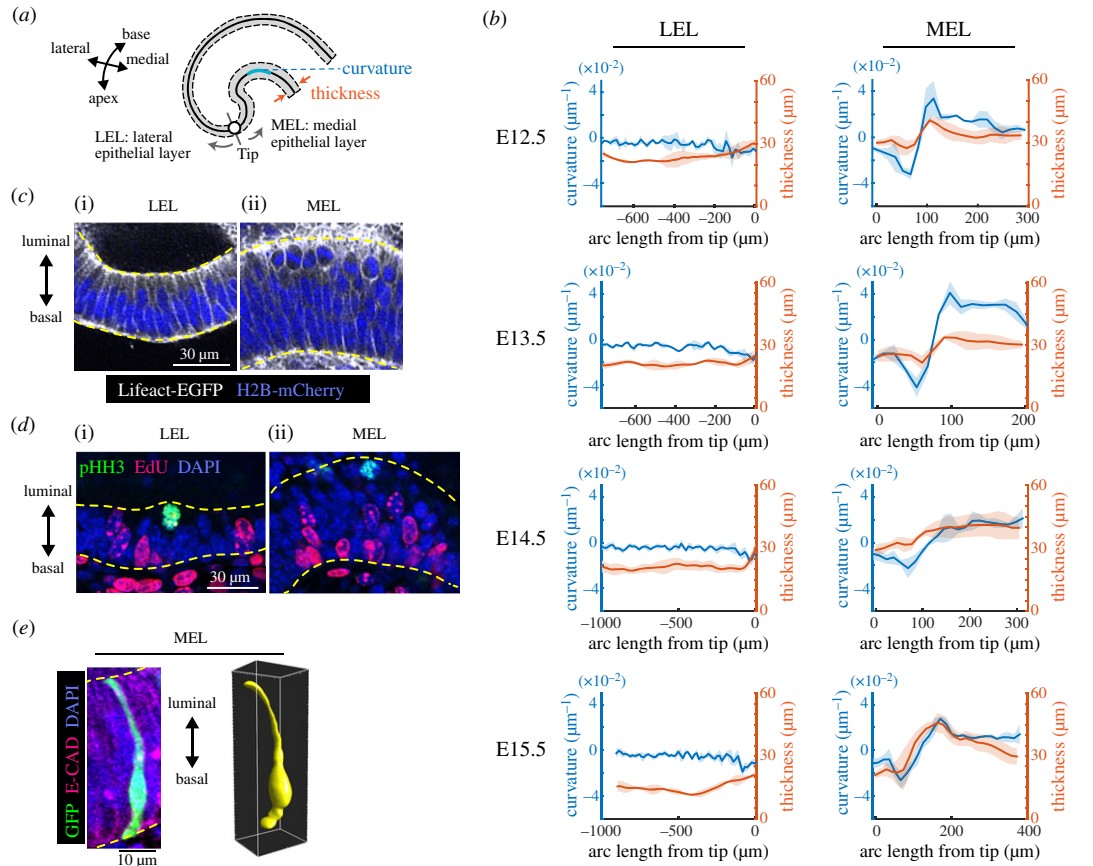

**Figure 2.** Morphological analysis shows the pseudostratified MEL curves in the cochlear duct. (*a*) Schematic diagram showing the regions used for morphological quantification. (*b*) Curvature and thickness as a function of the arc length from the apex tip along the LEL (left) and the MEL (right) between E12.5 and E15.5. Graphs describing the LEL from E12.5 to E14.5 were adopted from our previous study [30]. Values represent the mean ± standard deviation (s.d.) from $N = 3$. (*c*) Images of Lifeact-EGFP that labels F-actins (white) and H2B-mCherry (blue) at E13.5, showing cell shape and nuclear position in the LEL (i) and MEL (ii). Yellow dotted lines represent the luminal and basal edges. Scale bar = 30 μm. (*d*) Fluorescence labeling images describing anti-pHH3 (green), EdU (magenta) and nuclear DAPI counterstaining (blue) in the LEL (i) and MEL (ii) at E13.5. Yellow dotted lines represent the luminal and basal edges. Scale bar = 30 μm. (*e*) Fluorescence image of GFP-transfected cells labelling single cells (green) and anti-E-cadherin (magenta) with nuclear counterstaining using DAPI (blue) in the MEL (left) at E13.5 and the corresponding three-dimensional rendered image (right). Yellow dotted lines on the left represent the luminal and basal edges. Scale bar = 10 μm.

labyrinth was removed entirely to expose the cochlear duct prior to *ex vivo* culture. We used a reporter mouse line ubiquitously expressing fluorescent proteins localized to the cytoplasm, allowing us to identify the nuclei as regions without fluorescence, as well as the luminal and basal edges of the MEL in the pseudostratified epithelium. We observed differences in nuclear movement between the flat apex and curved base regions (figure 3*a*; electronic supplementary material, movie S2). The nuclei in the flat regions moved to the luminal edge of the epithelial layer before cell division, and the daughter nuclei returned to the basal edge after cell division (figure 3*b*). However, in the curved region, the cell nuclei remained on the luminal side of the cells even after cell division (figure 3*c*). To clarify this difference, we tracked the nuclei and quantified their distance from the luminal edge 3 h after nuclear division. After classifying the tissue into two categories, flat and curved, based on the half-maximal curvature value (figure 3*d*), we showed that fewer nuclei moved to the basal side of cells in the curved region than in the flat region (figure 3*d′*). This suggests that there are two different modes of luminal-basal nuclear movement in the MEL, depending on the distance from the apex tip. In addition, we examined two angles of cell division orientation, *φ* and *θ*, in the three-dimensional polar coordinates from the live imaging data, where *φ* and *θ* represent the zenith and azimuth angles, respectively (figure 3*e*). The *φ* angle was between 0° and 45°, indicating that cell division mostly occurred parallel to the luminal surface of the MEL (figure 3*f*), while the distribution of *θ* showed that most cells divided along with the apex-base axis rather than along with the roof-floor axis

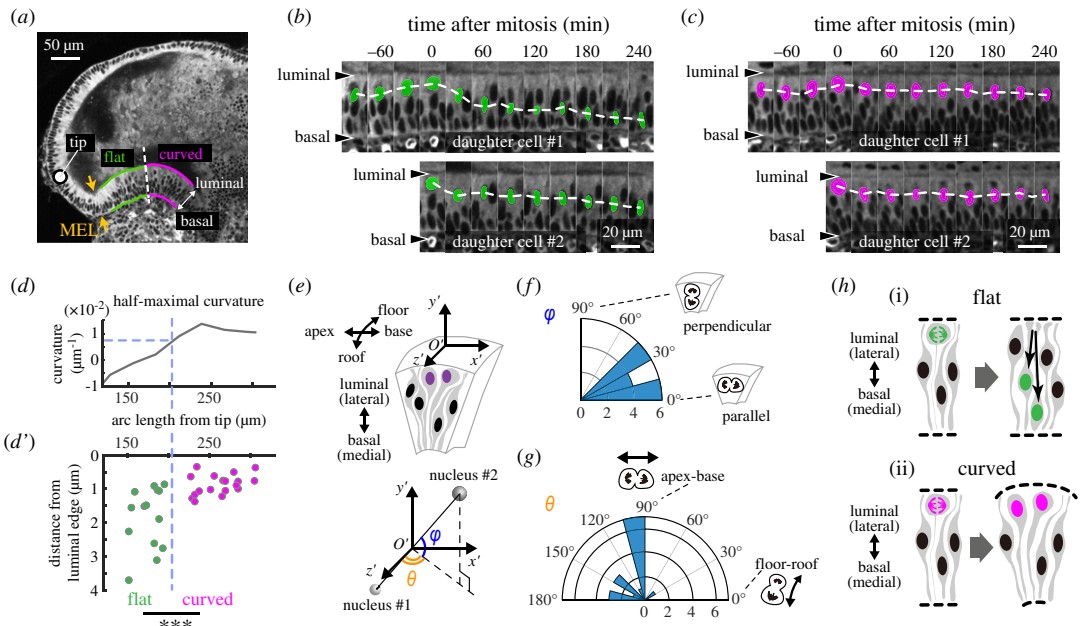

**Figure 3.** Live imaging shows that luminal nuclear stalling occurs in the curved portions of the pseudostratified epithelium. (a) Image of a section of an E14.5 cochlea in cytoplasmic reporter mice. The MEL was divided into flat (green) and curved (magenta) regions, and the boundary between the flat and curved regions is indicated by a dotted line. The circle indicates the apex duct tip. Scale bar = 50 μm. (b,c) Kymographic images of the flat (b) and curved regions (c). Coloured circles denote manually traced nuclei and dotted lines denote the change in position of the manually marked nuclei over time. Scale bars = 20 μm. (d,d′) We tracked 16 nuclear divisions and calculated their distance from the luminal edge 3 h after nuclear division and plotted these values against the arc length from the apex tip (d′) and corrected for the MEL curvature (d). We then used the half-maximal curvature (d) values to split the samples in (d′) into two groups, i.e. flat (n = 14) and curved (n = 18). The Mann–Whitney U test was used to evaluate statistical significance (p = 0.00038). (e) Nuclear division orientation is represented by two angles, φ and θ, in the polar coordinate O′. (f,g) Angle distribution of φ and θ. n = 16. Statistical significance was evaluated using a Rayleigh test revealing p = 0.00096 for (f) and p = 0.15 for (g). (h) Schematics describing the expected morphology based on the mode of nuclear movement. IKNM leads to flat morphologies (i) and asymmetric IKNM leads to curved morphologies (ii).

(figure 3g). Taken together, the data suggest that the orientation of cell division contributes to the apex-base localized expansion of the luminal side of the MEL.

Luminal-basal nuclear movement in the flat region exhibits interkinetic nuclear migration (IKNM) behaviours, which have been observed in various pseudostratified epithelial tissues [33–38]. During this process, nuclei at the basal side of the cell move closer to the luminal surface prior to mitosis, and the daughter nuclei move back to the basal side, which results in even growth within the epithelial layer (figure 3hi). This mode of nuclear migration does not necessarily generate bending of the epithelial tissue. By contrast, the lack of basalward movement during IKNM, which we refer to as 'luminal nuclear stalling', results in an asymmetric one-way flux of nuclei from the basal side of the tissue to the luminal surface, which eventually leads to local cell crowding and expansion due to increased division at the luminal side of the tissue (figure 3hii). Based on these observations, we hypothesize that the asymmetrical IKNM observed in the MEL and mediated by luminal nuclear stalling works in parallel with the oriented cell division to produce physical bending in the MEL via the differential growth of the luminal layers in these tissues.

## 2.4. Luminal nuclear stalling promotes medial epithelial layer bending

IKNM in pseudostratified tissues is dictated by the cytoskeletal machineries, including actomyosin and microtubules, during cell-cycle progression [35,36,38]. To examine the role of the cell cycle in IKNM, we perturbed the cell cycle using DNA synthesis inhibitor mitomycin C (MMC) and examined the morphological response of the MEL. As MMC prevents cells from entering the S-phase, IKNM can be halted before the nuclei arrive at the luminal side of the tissue. One day after treatment with 10 μM MMC E12.5 cochleae (figure 4a) presented with no EdU-positive S phase cells (figure 4b). This

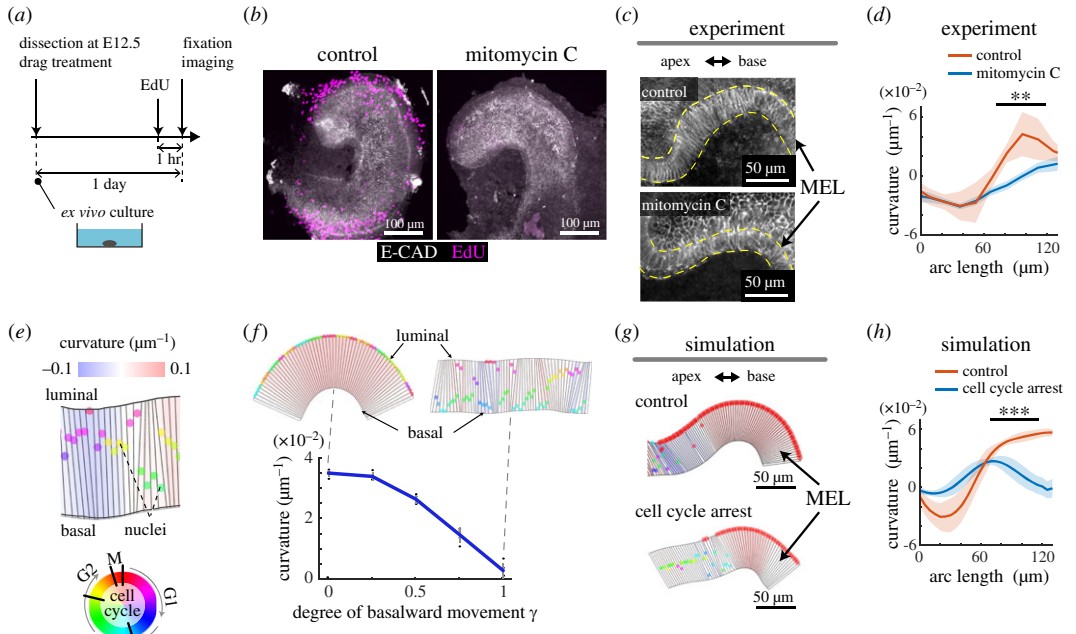

**Figure 4.** Pharmacological and mathematical analyses suggest that luminal nuclear stalling contributes to MEL bending via differential luminal-basal growth. (*a*) Schematics describing each of these experiments. (*b*) Representative images from the 10 µM MMC treatments of the cochlear tissues over a 24 h period. Maximum intensity projection of immunostained images for the anti-E-cadherin (white) and EdU labelling (magenta) are shown. Scale bars = 100 µm. (*c,d*) The morphological response to MMC treatment. Representative images of treated cochlear ducts visualized by immunostaining of anti-E-cadherin (*c*) and the measured curvature over the arc length (*d*). Values represent the mean ± s.d. from $N = 3$, and significance was evaluated using the Mann–Whitney U test, $p = 0.0017$. Scale bars = 50 µm. (*e*) Virtual epithelial tissues in the model. The coloured circles denote nuclei with the cell cycle state represented at the bottom. The tissue curvature is represented in the change from blue-to-red as shown on the top of the image. (*f*) Numerical investigation describing tissue curvature as a measure of basalward movement $\gamma$. Values represent the mean ± s.d. from $N = 10$. (*g,h*) Model prediction following cell cycle arrest before S phase entry in the growing virtual epithelial tissues. Representative images (*g*) and the curvature over the arc length (*h*). Values represent the mean ± s.d. from $N = 10$ with statistical significance evaluated using the Mann–Whitney U test, $p < 0.001$. Scale bars = 50 µm.

resulted in a marked decrease in the curvature along with the MEL, approximately 100 µm away from the apex tip, but not around the tip (figure 4*c,d*), suggesting that MEL bending is promoted by cell-cycle-dependent events.

Finally, we built an agent-based mechanical model and examined whether cell-cycle-dependent nuclear movements affect the curvature of a virtual epithelial tissue. Epithelial cells were represented as polygons with vertices consisting of the luminal and basal edges, which could deform according to the cell-cycle-dependent nuclear position, based on mechanical interactions between neighbouring cells (figure 4*e*). In this model, each polygon was set to occupy a certain space on the basal side, even when the nucleus in the polygon was at the luminal edge. Thus, a single polygon represents averaged cellular dynamics at a position along with the apex-base axis rather than representing a single cell. We then introduced parameter $\gamma$ controlling the degree of basalward movement after IKNM—the nucleus moves to the basal edge when $\gamma = 1$ and stays at the luminal side when $\gamma = 0$. Our mathematical simulation demonstrated that the curvature of the epithelial tissue monotonically decreased as a function of $\gamma$ (figure 4*f*), which supports our observation that luminal nuclear stalling contributes to the mechanical bending of the epithelial tissue. Moreover, we performed *in silico* experiments with measured and tuned parameters from the obtained images and numerical simulations (electronic supplementary material, figure S1 and Methods), and found that cell cycle arrest at S-phase entry decreased the curvature of the MEL (figure 4*g,h*; electronic supplementary material, movie S3). This model simulation was used to demonstrate the plausibility of this concept in an ideal situation, rather than precisely incorporating realistic parameters. Thus, the nuclear positions were mostly localized at the luminal side in the control case, resulting in differential growth between the luminal and basal sides. By contrast, simulations of the cell cycle arrest showed that the nuclear positions were dispersed within the virtual epithelial tissue, which eventually mitigated tissue bending.

# 3. Discussion

Here, we have used two-photon excitation microscopy to observe the cellular dynamics in the MEL of developing cochlear ducts under *ex vivo* culture conditions. Our deep tissue imaging has illuminated the unprecedented dynamics of these cells which underpin the physical basis for cochlear duct bending; that is, the movement of the nuclei from the basal side of the tissue to the luminal side of the tissue and their stalling in this location following cell division, specifically in the curved region. Moreover, when we combined this data with our simulations of the mathematical model, we have demonstrated that luminal nuclear stalling would give rise to the bending of the pseudostratified tissue via differential tissue growth between the luminal and basal sides of the MEL. Given this, we propose that interkinetic luminal nuclear stalling is a driver for the bending morphogenesis in the murine cochlear duct.

We showed that the mode of luminal-basal nuclear migration switches is based on position in the continuous pseudostratified epithelium (figure 3*d,d'*), which controls the curvature of the MEL. This dynamic process is observed during the developmental process, but not in the adult. To the best of our knowledge, this is the first study to describe nuclei stalling on the luminal side of the pseudostratified epithelium during IKNM in the normal development of murine cochlear duct. Moreover, our results support the previously proposed idea that alteration of the cell-cycle phase during IKNM could regulate the bending of the chick neural plate during development [39]. One interesting possibility is that this luminal stalling might be the result of a nuclear density gradient along with the luminal-basal axis provided by the convexity of the luminal side and physical interactions between various other properties, including nuclear movement, tissue mechanics and geometry. Although it remains unclear whether convexity causes luminal nuclear stalling, luminal nuclear stalling results in some luminal expansion to allow for the occupation of the physical space, even in the curved pseudo stratified epithelium. Therefore, we propose that the luminal expansion driven by luminal nuclear stalling is a possible mechanism for sculpting curves in the pseudostratified epithelium and that this may act in concert with the actomyosin-based basal shrinkage that has been reported in zebrafish neuroepithelium [40,41].

One more important question remains: What causes the change in the nuclear movement of a specific region along with the apex-base axis on the medial side (figure 3*d,d'*)? Previous evaluations have shown that TAG-1 (transient axonal glycoprotein-1) knockdown impedes basalward nuclear movement during IKNM in the ventricular zone of murine embryonic brains, leading to overcrowding of the neural progenitor cells on the luminal side of the structures and severe cortical dysplasia [42]. It has also been shown that cytoskeletal machinery, including actomyosin and microtubules, regulates basalward nuclear movement [35,36,38]. These observations provide the basis for the identification of the molecules responsible for nuclear behaviour in the MEL, which need to be clarified in future studies. In addition, the questions of which unknown chemical and mechanical factors provide positional information for the change in nuclear behaviours and how each interacts with these complex systems still remain. The region of luminal nuclear stalling would reflect the morphological profiles in curvature and thickness of the MEL. Therefore, based on our results (figure 2*b*), we infer that the region of luminal nuclear stalling is continuously updated at a constant distance away from the growing apex tip of the cochlear duct.

Although we propose luminal-basal differential growth caused by luminal nuclear stalling as a physical mechanism for MEL bending in the cochlear duct, further experimental validation is required. This is highlighted by the main limitation of this study, the lack of a method to manipulate nuclear and cellular behaviours in specific regions of the MEL. Controlling IKNM by manipulating the cytoskeletal elements in specific regions of the MEL is far more challenging. Instead, we partially arrested the cell cycle in the cochleae using MMC. However, this experiment may be flawed as cell proliferation was halted throughout the explant, including the mesenchyme and spiral ganglions. To overcome this problem, spatio-temporal-specific perturbations of the cytoskeletal machinery using optogenetics and photoactivatable pharmacological agents may provide a better understanding of these complex multicellular interactions [43–47].

In summary, we have reported that cellular nuclei stall at the luminal side of the pseudostratified epithelium after cell division in the curved portions of the cochlea but return to the basal side in the flat epithelium during murine cochlear duct development. We have also suggested that luminal nuclear stalling could be a physical factor in the differential growth of the MEL during cochlear duct development. Note that stalling IKNM would contribute to the bending but not to the torsion of the spiral formation. The torsional force can be generated by the helical cellular flows from the base-floor

region to the apex-roof region on the lateral side of the developing cochlear duct [30,48]. We believe that the interplay between the medial nuclear stalling reported in this study and this lateral cellular flow underlies the spiral morphogenesis of the cochlear duct.

# 4. Material and methods

## 4.1. Experiments

### 4.1.1. Animals

We used the transgenic mice that ubiquitously express a CFP variant and a YFP variant localized in the cytosol reported elsewhere [30]. For simultaneous imaging of F-actin and nuclei, we crossed Lifeact-EGFP [49] and R26-H2B-mCherry [50]. Lifeact-EGFP mice were generously provided by Takashi Hiiragi from EMBL Heidelberg and R26-H2B-mCherry mice were provided from RIKEN Large (CDB0204 K). Otherwise, we used ICR mice purchased from Japan SLC, Inc. We designated the midnight preceding the plug as embryonic day 0.0 (E0.0), and all mice were sacrificed by cervical dislocation to minimize suffering. All the animal experiments were approved by the local ethical committee for animal experimentation (MedKyo 19090 and 20081) and were performed in compliance with the guide for the care and use of laboratory animals at Kyoto University.

### 4.1.2. Antibodies and small-molecule inhibitors

The following primary and secondary antibodies were used for immunofluorescence: anti-E-cadherin rat antibody (Thermo Fisher Scientific, 13-1900, 1 : 100 dilution), anti-Myosin-VIIa rabbit polyclonal antibody (Proteus BioSciences Inc., #25-6790, 1 : 200 dilution), anti-Histone H3 (phospho S28) rat polyclonal antibody (Abcam, #ab10543, 1 : 200 dilution), Alexa Fluor 546-conjugated goat anti-rat IgG (H + L) antibody (Thermo Fisher Scientific, #A11081, 1 : 1000 dilution) and Alexa Fluor 647-conjugated goat anti-rabbit IgG (H + L) antibody (Abcam, #ab150079, or Thermo Fisher Scientific, #A21247, 1 : 1000 dilution). We used MMC (Nacalai Tesque, #20898-2).

### 4.1.3. Surgical separation

We dissected cochleae from embryos at E17.5 and manually separated a cochlear duct into the medial and lateral side. The images and movies were acquired using the stereomicroscope (SZX16, Olympus) with a cooled colour CCD camera (DP73, Olympus).

### 4.1.4. Explant cultures

Murine cochleae were dissected without removing the capsule unless otherwise noted. The dissected cochleae were put on a 35 mm glass-based dish (Iwaki, #3910-035), and they were mounted with 1 µl of growth factor reduced Matrigel (Corning, #356231), filled with 2 ml of a culture medium including FluoroBrite DMEM Media (Thermo Fischer Scientific, #A1896701), 1% GlutaMAX (Thermo Fischer Scientific, #35050061) and 1% N2 Supplement with Transferrin (Holo)(FUJIFILM Wako Pure Chemical Corporation, #141-08941) for the culture at 37°C under 5% $CO_2$.

### 4.1.5. Mosaic cell labelling

DNA solutions of the pCAG-GFP vectors (0.7 µg µl$^{-1}$) in PBS with 0.1% Fast Green FCF (Sigma-Aldrich, #F7252-5G) were injected into the lumen of an inner ear dissected from E13.5 embryos through a fine glass capillary tube under a stereomicroscope (SZ61, Olympus). After injection of DNA solution, the cochlea was sandwiched by a pair of tweezer type electrodes (Nepa Gene Co., #CUY650P5) and the DNA was electroporated using the NEPA21 Super Electroporator (Nepa Gene Co.). Three 5 ms square pulses (175 V, +) with a 10% decay rate at intervals of 50 ms were applied as poring pulses, followed by three 50 ms square pulses (10 V, +/−) with a 40% decay rate at intervals of 50 ms as transfer pulses.

### 4.1.6. Fluorescence staining

For the whole-tissue staining, we first removed cartilaginous capsule from the dissected cochleae and performed the fluorescence staining according to the previous study [51]. Briefly, the samples were

fixed with 4% PFA in PBS overnight at 4°C and then blocked by incubation in 10% normal goat serum (Abcam, #ab156046) diluted in 0.1% Triton X-100/PBS (PBT) for 3 h at 37°C. The samples were treated with primary antibodies overnight at 4°C, washed in 0.1% PBT and subsequently treated with secondary antibodies conjugated to either Alexa Fluor 546 or Alexa Fluor 647 overnight at 4°C. DAPI (Dojindo Molecular Technologies, #D523-10, 1 : 200 dilution) was used for counterstaining of nucleus.

### 4.1.7. Ethynyl deoxyuridine assay

For the EdU incorporation to dissected cochleae, we added 10 µM EdU dissolved into PBS to the culture media 1 h prior to the fixation. Next, whole-tissue immunostaining of E-cadherin and nuclear counterstaining with DAPI were performed as described above. Then, EdU signal was detected using the Click-iT® EdU Imaging Kits (Thermo Fisher Scientific, #C10340).

### 4.1.8. Fluorescence imaging

For live imaging of MEL, we completely removed the cochlear capsule and put the intact cochlear duct attached with spiral ganglion onto the glass-bottom dish whose apex-floor side faces the glass. For long-term organ-scale imaging, used previously [30] but not in this study, we partially cut off the capsule adjacent to the apex tip of cochlear ducts using tweezers carefully, and the semicircular canals were removed. The former method allows us to take images of MEL at single-cell resolution but failed to achieve elongation and bending of the cochlear duct. By contrast, the latter method recapitulates the cochlear duct morphogenesis but was unable to detect fluorescence signals in the medial side. We performed time-lapse microscopy using an incubator-integrated multiphoton fluorescence microscope system (LCV-MPE, Olympus) with a ×25 water-immersion lens (NA = 1.05, WD = 2 mm, XLPLN25XWMP2, Olympus) or an inverted microscope system (FV1200MPE-IX83, Olympus) with a ×30 silicone-immersion lens (NA = 1.05, WD = 0.8 mm, UPLSAPO30XS, Olympus). The excitation wavelengths were set to 840 nm, 930 nm and 1040 nm each for CFP, Lifeact-EGFP, and R26-H2B-mCherry (InSight DeepSee, Spectra-Physics). Imaging conditions for Lifeact-EGFP and R26-H2B-mCherry were as follows—scan size: 800 × 800 pixels, scan speed: 4.0 µsec pixel$^{-1}$, IR cut filter: RDM690 (Olympus), dichroic mirrors: DM505 and DM570 (Olympus), and emission filters: BA495–540 for EGFP and BA575–630 for mCherry (Olympus). Imaging conditions for the CFP were as follows—scan size: 800 × 800 pixels, scan speed: 10 µsec pixel$^{-1}$, IR cut filter: RDM690 (Olympus), dichroic mirrors: DM505 and DM570 (Olympus), and emission filters: BA460–500 (Olympus).

For fixed samples stained with dyes and antibodies, we first covered the samples with 10 µl of 1% agarose gel onto a glass-based dish (Greiner Bio-One, #627871). Next, the samples were optically cleared by immerging with the BABB (benzyl-alcohol and benzyl-benzoate, 1:2, #04520-96 and # 04601-65, Nacalai Tesque) solution or CUBIC-R+ (Tokyo Chemical Industry Co., # T3741) solution. Stack images were then acquired using the confocal laser scanning platform Leica TCS SP8 equipped with the hybrid detector Leica HyD with the ×40 objective lens (NA = 1.3, WD = 240 µm, HC PL APO CS2, Leica) and the Olympus FluoView FV1000 with the ×30 objective lens (NA = 1.05, WD = 0.8 mm, UPLSAPO30XS, Olympus).

## 4.2. Quantification and analysis

### 4.2.1. Measurement of layer curvature and thickness

For two-dimensional measurement of curvature and thickness, we first performed whole-mount immunofluorescence of E-cadherin to visualize the cochlear epithelium and acquired z-stack images by confocal microscopy as described above. Next, we manually traced the luminal and basal side of epithelial cells on the middle horizontal section of the roof-floor axis. The extracted epithelial layer was named as the MEL/LEL according to the side based on a manually chosen apex tip point. Then, the curve of the epithelial layer was obtained by the iterative skeletonization, and discrete points $(x_i, y_i)$ were sampled along with the curves at regular intervals of 15 µm. Finally, fitting the discrete points with a cubic spline function, the function $S_i$ at an interval $[x_i, x_{i+1}]$ is denoted as

$$S_i(x) = a_i(x - x_i)^3 + b_i(x - x_i)^2 + c_i(x - x_i) + d_i.$$

Due to a definition of curvature $\kappa(x) = S''(1 + S'^2)^{-3/2}$, the curvature from the spline function was calculated as

$$\kappa_i(x) = \frac{6a(x - x_i) + 2b}{\left(1 + \{3a(x - x_i)^2 + 2b(x - x_i) + c\}^2\right)^{3/2}}.$$

The curve of the convex/concave to the duct lumen was assigned as positive/negative in $\kappa$. We defined the layer thickness as a linear length connecting to the luminal and basal edge, which is vertical to the curve of the epithelial layer at sampling points.

### 4.2.2. Quantification of nuclear migration

We manually measured the centre position of a daughter cell and its luminal and basal edge. Based on the cross point where the luminal-basal line and the centre line of MEL are intersected, we calculated the arc length from the duct tip to the nuclear position using a custom-built MATLAB code.

### 4.2.3. Angle measurement of cell division orientation

First, we performed two-photon live imaging using FV1200MPE-IX83 as described above and obtained volumetric images with a depth interval of 3 µm. In each mitotic event, we then manually measured position (x, y, z) of the two daughter cells 3 h later after the mitosis in the fixed coordinate system O of the three-dimensional image. Also, we manually acquired the position of the luminal and basal edge of the mother cell to define a position (x′, y′, z′) in a local coordinate system, O′, the origin of which is the middle point of the two daughter nuclei. Each orthogonal basis of the O′ coordinate was defined as follows: the apex-base axis in the MEL (x′), the surface normal of the MEL (y′), i.e. luminal-basal axis in the MEL and roof-floor axis (z′) orthogonal to both x′ and y′ according to the right-handed system. With this local coordinate system, we finally calculated two angles $\varphi$ and $\theta$ in the local sphere coordinate resulting from coordinate transformation from the O into the O′ system.

### 4.2.4. Statistical analysis

The number of cells or region of interests analysed ($n$) and the number of biological replicates ($N$) are indicated in the figure legends. No particular statistical method was used to predetermine the sample size. A minimum of $N = 3$ independent experiments was performed. Statistical tests, sample sizes, test statistics and $p$-values were described in the main text. $p$-values of less than 0.05 were considered to be statistically significant in two-tailed tests and were classified as four categories: ***($p < 0.001$), **($p < 0.01$), *($p < 0.05$) and n.s. (not significant, i.e. $p \geq 0.05$).

### 4.2.5. Software

For digital image processing, we used MATLAB (MathWorks) and Image J (National Institute of Health). For graphics, we used MATLAB (MathWorks), Imaris (Bitplane) and ImageJ (National Institute of Health). For statistical analysis, we used MATLAB (MathWorks).

## 4.3. Mathematical model

### 4.3.1. A mechanical model for medial epithelial layer morphogenesis

We modelled multicellular dynamics within an epithelial layer using a vertex dynamics model (VDM) [52–54]. Here, we focused on a two-dimensional section on apex-base and roof-floor axes. In general, the two-dimensional VDM model represents a single cell as a polygon, of which vertices are elementary points that constitute the cell shape, and a group of cells is regarded as a set of polygons shared by neighbouring cells. However, in this model, each polygon was set to occupy with the certain size at the basal side even when the nucleus in the polygon was at the luminal side. Thus, single polygon represents averaged cellular dynamics at a position along with the apex-base axis rather than representing a single cell as described in the main text. We put four vertices, two of which are regarded as a luminal edge and the other two of which are as a basal edge.

In the VDM, the dynamics of position of vertex $i$, $r_i$, obey the equation of motion based on the principle of least potential energy $U$ as follows:

$$\eta(\dot{r}_i - v_i) = -\nabla_i U \tag{4.1}$$

where $\eta$ is a viscosity coefficient. $v_i$ is a local velocity of vertex $i$, defined as $\langle v_i \rangle = \langle \dot{r}_{j_i} \rangle_{j_i}$, where $j_i$ is an index of cells contacting to vertex $i$, $r_{j_i}$ is a centroid of cell $j_i$ and $\langle \rangle_{j_i}$ denotes averaging in $j_i$ [54–56]. For the potential energy as a minimum expression, we defined as

$$U = \sum_j \left\{ \frac{k_a}{2}(a_j - a_j^*)^2 + \frac{k_b}{2}(b_j - b_j^*)^2 + \frac{k_l}{2}(l_j - l_j^*)^2 + \frac{k_A}{2}(A_j - A_j^*)^2 \right\} + \sum_i \frac{k_\theta}{2}\theta_i^2. \tag{4.2}$$

The first to third terms each represent a regulation of cell edge length in luminal/apical, basal and lateral side with controlling parameters ($k_a$, $k_b$, $k_l$), current length ($a_j$, $b_j$, $l_j$) and the target length ($a_j^*$, $b_j^*$, $l_j^*$). Target edge length of the luminal/apical and that of basal side are a function of nuclear position within the epithelial layer, described later. The fourth term represents cell area preservation with its coefficient $k_A$, current cell area $A_j$ and the target cell area $A_j^*$. The fifth term represents bending energy of the luminal/apical side and that of basal side attributed to each vertex; $k_\theta$ denotes the bending rigidity and $\theta_i$ is an angle of luminal/apical edge or that of basal edge at a vertex $i$. Although the cell shape is relatively flexible in a pseudostratified epithelial tissue, this model framework is valid as several biological features can be incorporated, such as the nuclear position-dependent edge regulation and cell size preservation.

Each cell has a unique cell cycle, length of which $\tau_{\mathrm{div}}$ is assumed to be equally assigned to all cells. We assume that a period of each cell cycle phase (G1, S, G2, and M) is partitioned as $11:8:4:1$, and a cell cycle status or timer $\tau_j$, origin of which is defined at the end of cell division, i.e. a boundary of M-G1, is provided according to the distribution of cell cycle phase. We model that the cell timer $\tau_j$ in the cycle determines the distance from the luminal side of a cell, i.e. nuclear position $d_j$ ($0 \leq d_j \leq 1$), with a parameter $\gamma$ as follows:

$$d_j = \begin{cases} \dfrac{2\gamma\tau_j}{\tau_{\mathrm{div}}} & (0 < \tau_j \leq 0.5\tau_{\mathrm{div}}) \\ \dfrac{(-\tau_j/\tau_{\mathrm{div}} + 0.95)\gamma}{0.45} & (0.5\tau_{\mathrm{div}} < \tau_j \leq 0.95\tau_{\mathrm{div}}) \\ 0 & (0.95\tau_{\mathrm{div}} < \tau_j \leq \tau_{\mathrm{div}}). \end{cases} \tag{4.3}$$

This means that nuclei move toward the basal side of cell during G1-S phase with a degree $\gamma$ and get close to the luminal side before entering M phase. Of note, $\gamma$ controls the degree of basalward movement as described in the main text. Once a cell divides, $\tau_j$ in one of daughter cells reset to zero and one in the other is set to a value stochastically chosen from 0 to $0.1\tau_{\mathrm{div}}$ with a uniform distribution to avoid perfect synchronization between neighbouring cells. Also, the initial cell phase was chosen at random.

Occupied cell area is dominant in the MEL and the cell position along with the luminal-basal axis should regulate the length of luminal edge and that of basal edge. Thus, we define the target length as the simplest linear function of the nuclear position with two parameters $\xi_{\max}$ and $\xi_{\min}$ as follows:

$$\begin{aligned} a_j^* &= (\xi_{\min} - \xi_{\max})d_j + \xi_{\max} \\ b_j^* &= (\xi_{\max} - \xi_{\min})d_j + \xi_{\min}. \end{aligned} \tag{4.4}$$

It should be emphasized that we did not aim to incorporate the physical situation precisely but rather used it to demonstrate the concept in some ideal situations. Here, we used the vertex modelling framework, but another possible approach to express individual nuclear dynamics was a particle-based model, which would ignore the cell shape and focus only on the centre of mass of the nuclei. For this model, it would not be easy to link cellular dynamics and mechanical integrity of the epithelial tissues with densely packed nuclei—this is a challenge for future studies.

### 4.3.2. Numerical simulation

The ordinary differential equations were numerically solved by the forward Euler method with time step 0.01. The code was generated with MATLAB (MathWorks). Regarding the initial condition, 20 rectangular polygons, width/height of each which is 2.5/50 μm, are arrayed along a horizontal line. Standard parameter set is $\eta = 1$, $k_A = 0.01$, $a_j^* = 125$ [$\mu m^2$], $k_a = 1$, $k_b = 1$, $k_l = 1$, $k_\theta = 30$, $l_j^* = 50$ [$\mu m^2$], $\tau_{\mathrm{div}} = 432$, $\gamma = 0.9$, $\xi_{\max} = 5$ [μm], $\xi_{\min} = 1$ [μm], otherwise noted. $a_j^*$, $l_j^*$, $\xi_{\max}$ and $\xi_{\min}$ were determined

from obtained images. We set the value of $\gamma$ because the nuclei fully returned to the basal side according to the cell cycle length in the simulation, which occurred only in the uncurved region of the MEL. Other parameters were first chosen empirically and the plausibility for the numerical simulation was tested by fitting with the experimental data on the MEL curvature. We demonstrated the similarity of the simulated results to the experimental ones by introducing the root-mean-square error (RMSE), to measure the differences in datasets between simulations and experiments. The RMSE at the standard parameter set showed a minimal value and small variance. Note that relative values rather than absolute ones are critical for the dynamics.

In the simulations, the edges of the cells at the tissue boundary were set to be free within the finite window size, which was determined as a practical condition for the quantitative investigation. By contrast, under experimental conditions, the boundary cells were constrained by other cells outside the corresponding window. We were concerned that these different boundary conditions caused the quantitative difference in the curves between experiments and simulations.

### 4.3.3. Detailed setting in virtual experiments for cell cycle arrest

For an investigation towards mimicking the developmental process, we introduced luminal stalling region, defined with an arc length along with the epithelial layer $L$ originated from the boundary of apex side. Assuming once the cells in the luminal stalling region undergo division, the daughter cells' timer $\tau_j$ do not count up, meaning that those nuclei stay at the luminal side without reentering the cell cycle. This is due to experimental observations in which daughter cells do not undergo mitosis exclusively in the curved region within a short time scale, at least our observation window. For the cells in the non-luminal stalling region, $\tau_{\text{div}}$ was set stochastically chosen from a uniform distribution from 216 to 864 each after the cell division to incorporate variability and asynchronicity in nuclear dynamics. To recapitulate the MMC treatment in simulation, the cell cycle was arrested if the cell timer $\tau_j$ was ranged in $0.25\tau_{\text{div}}$ to $0.8\tau_{\text{div}}$, corresponding to the mid G1 phase to the end of S phase, when the simulation time exceeded 700. For the control case, we evaluated the curvature of virtual MEL when the cell number reached at 100. For the case of MMC treatment, we evaluated when the simulation time reached at 1300 because the averaged simulation time in the control case was 1298 ($n = 10$).

Data accessibility. Relevant codes for this research work are stored in GitHub: https://github.com/tsuyoshihirashima/cochlea_vertex.git and have been archived within the Zenodo repository: https://zenodo.org/record/5636659#.YYAF89axiCQ [57].

The data are provided in the electronic supplementary material [58].

Authors' contributions. Conceptualization was carried out by M.I. and T.H.; methodology was carried out by M.I., T.T., M.M. and T.H.; software was carried out by M.I. and T.H.; validation was carried out by M.I. and T.H.; formal analysis was done by M.I. and T.H.; investigation was carried out by M.I. and T.H.; resources was carried out by M.I., M.M. and T.H.; data curation was carried out by M.I. and T.H.; writing the original draft was carried out by M.I. and T.H.; writing the review and editing was carried out by M.I., T.T., M.M. and T.H.; visualization was carried out by M.I. and T.H.; supervision was carried out by M.M. and T.H.; project administration was carried out by T.H.; funding acquisition was carried out by M.M. and T.H.

Competing interests. The authors declare no competing interests.

Funding. This work was supported by JST PRESTO JPMJPR1949, by JSPS KAKENHI 21H05290, by Narishige Neuroscience Research Foundation and by CREST JPMJCR1654.

Acknowledgements. We would like to thank Akane Kusumi and Yu Kurata for technical assistance and Yoshiko Takahashi for fruitful discussions. This work was supported by Kyoto University Live Imaging Center.

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
