## [Peer Review File · Royal Society Open Science]

Review History

RSOS-211024.R0 (Original submission)

Review form: Reviewer 1 (Raj Ladher)

Is the manuscript scientifically sound in its present form?

Yes

Are the interpretations and conclusions justified by the results?

Yes

Is the language acceptable?

Yes

Do you have any ethical concerns with this paper?

No

Have you any concerns about statistical analyses in this paper?

No

Recommendation?

Accept with minor revision (please list in comments)

Comments to the Author(s)

The study from Ishii et al., suggests that nuclear stalling may play a role in the bending of the cochlear duct in mice. The study is beautiful, and the model presented interesting. One possible inclusion would be what accounting for the no of turns in the duct. Are all the turns points of luminal nuclear stalling? This would be a discussion point, but it would be interesting to think about the borders between the flat and curved parts of the medial epithelia particularly at E16.5 - E18.5 when their are more turns.

Another possible inclusion would be a discussion about pitch - nuclear stalling cannot really explain pitch, or there other mechanisms that may account for this?

Review form: Reviewer 2**Is the manuscript scientifically sound in its present form?**

Yes

Are the interpretations and conclusions justified by the results?

Yes

Is the language acceptable?

Yes

Do you have any ethical concerns with this paper?

No

Have you any concerns about statistical analyses in this paper?

No

Recommendation?

Accept with minor revision (please list in comments)

Comments to the Author(s)

(1)

The statement is interesting that the luminal nuclear stalling in the interkinetic nuclear migration is related to morphogenesis of tissues. However, I am wondering how long the luminal nuclear stalling continues. Is it temporary behavior during morphogenesis of developing cochlea? How about is it observed in the cochlea of adult animals? If the authors have any data or suggestive information about this question, they are expected to describe.

(2)

In morphogenesis of the cochlea, the authors described that, for elongation of the cochlear duct, sonic hedgehog signaling, fibroblast growth factor signaling and non-canonical Wnt-planar cell polarity signaling were required. In addition, they elucidated that the luminal nuclear stalling caused cochlear duct bending. On the other hand, animals have a pair of the cochleae, which are mirror image with each other. What defines chirality of the cochleae of left and right sides of the body. I would like to hear what the authors think about this problem?

Decision letter (RSOS-211024.R0)

Dear Mr Ishii

On behalf of the Editors, we are pleased to inform you that your Manuscript RSOS-211024 "Stalling interkinetic nuclear migration in curved pseudostratified epithelium of developing cochlea" has been accepted for publication in Royal Society Open Science subject to minor revision in accordance with the referees' reports. Please find the referees' comments along with any feedback from the Editors below my signature.

Please submit your revised manuscript and required files (see below) no later than 7 days from today's (ie 22-Oct-2021) date. Note: the ScholarOne system will 'lock' if submission of the revision is attempted 7 or more days after the deadline. If you do not think you will be able to meet this deadline please contact the editorial office immediately.

on behalf of Professor Sebastian Shimeld (Associate Editor) and Pietro Cicuta (Subject Editor)
openscience@royalsociety.org

Associate Editor Comments to Author (Professor Sebastian Shimeld):
Comments to the Author:
Dear Professor Hirashima

Thank you for your patience in waiting for these comments, and I apologise for the length of time the process has taken. Your manuscript has now been seen by two expert reviewers, both of whom are broadly supportive of your work. Both also suggest a small number of changes that they feel will help improve the manuscript, essentially to the discussion section where they ask you to develop some of the ideas and implications a little further. I agree with the reviewers, and think this wider discussion would improve and already interesting manuscript and broaden its relevance and implications.

With best wishes
Seb Shimeld

Reviewer comments to Author:

Reviewer: 1

Comments to the Author(s)

The study from Ishii et al., suggests that nuclear stalling may play a role in the bending of the cochlear duct in mice. The study is beautiful, and the model presented interesting. One possible inclusion would be what accounting for the no of turns in the duct. Are all the turns points of luminal nuclear stalling? This would be a discussion point, but it would be interesting to think about the borders between the flat and curved parts of the medial epithelia particularly at E16.5 - E18.5 when there are more turns.

Another possible inclusion would be a discussion about pitch - nuclear stalling cannot really explain pitch, or there other mechanisms that may account for this?

Reviewer: 2

Comments to the Author(s)

(1)

The statement is interesting that the luminal nuclear stalling in the interkinetic nuclear migration is related to morphogenesis of tissues. However, I am wondering how long the luminal nuclear stalling continues. Is it temporary behavior during morphogenesis of developing cochlea? How about is it observed in the cochlea of adult animals? If the authors have any data or suggestive information about this question, they are expected to describe.

(2)

In morphogenesis of the cochlea, the authors described that, for elongation of the cochlear duct, sonic hedgehog signaling, fibroblast growth factor signaling and non-canonical Wnt-planar cell polarity signaling were required. In addition, they elucidated that the luminal nuclear stalling caused cochlear duct bending. On the other hand, animals have a pair of the cochleae, which are mirror image with each other. What defines chirality of the cochleae of left and right sides of the body. I would like to hear what the authors think about this problem?

===PREPARING YOUR MANUSCRIPT===

one version should clearly identify all the changes that have been made (for instance, in coloured highlight, in bold text, or tracked changes);

===PREPARING YOUR REVISION IN SCHOLARONE===

- Ensure that your data access statement meets the requirements at <https://royalsociety.org/journals/authors/author-guidelines/#data>. You should ensure that you cite the dataset in your reference list. If you have deposited data etc in the Dryad repository, please only include the 'For publication' link at this stage. You should remove the 'For review' link.
- If you are requesting an article processing charge waiver, you must select the relevant waiver option (if requesting a discretionary waiver, the form should have been uploaded, see 'File upload' above).
- If you have uploaded any electronic supplementary (ESM) files, please ensure you follow the guidance at <https://royalsociety.org/journals/authors/author-guidelines/#supplementary-material> to include a suitable title and informative caption. An example of appropriate titling and captioning may be found at https://figshare.com/articles/Table_S2_from_Is_there_a_trade-off_between_peak_performance_and_performance_breadth_across_temperatures_for_aerobic_scope_in_teleost_fishes_/3843624.

Author's Response to Decision Letter for (RSOS-211024.R0)

See Appendix A.

Decision letter (RSOS-211024.R1)

Dear Mr Ishii,

I am pleased to inform you that your manuscript entitled "Stalling interkinetic nuclear migration in curved pseudostratified epithelium of developing cochlea" is now accepted for publication in Royal Society Open Science.

The proof of your paper will be available for review using the Royal Society online proofing system and you will receive details of how to access this in the near future from our production office (openscience_proofs@royalsociety.org). We aim to maintain rapid times to publication after

acceptance of your manuscript and we would ask you to please contact both the production office and editorial office if you are likely to be away from e-mail contact to minimise delays to publication. If you are going to be away, please nominate a co-author (if available) to manage the proofing process, and ensure they are copied into your email to the journal.

on behalf of Professor Sebastian Shimeld (Associate Editor) and Pietro Cicuta (Subject Editor)
openscience@royalsociety.org

Appendix A

Replies to the comments: RSOS-211024

“Stalling interkinetic nuclear migration in curved pseudostratified epithelium of developing cochlea”

We would like to thank the reviewers for their thoughtful responses and efforts toward improving our manuscript. After carefully reading the reviewers’ comments, we have responded to all concerns and revised the manuscript according to the suggestions. A point-by-point response to each reviewer’s concerns is provided below. For convenience, we have included the reviewers’ comments in italics below, followed by our replies.

Reviewer: 1

The study from Ishii et al., suggests that nuclear stalling may play a role in the bending of the cochlear duct in mice. The study is beautiful, and the model presented interesting. One possible inclusion would be what accounting for the no of turns in the duct. Are all the turns points of luminal nuclear stalling? This would be a discussion point, but it would be interesting to think about the borders between the flat and curved parts of the medial epithelia particularly at E16.5 - E18.5 when there are more turns.

Thank you for your comments. We’ve considered that the turning points would not be determined as a discrete process but generated continuously through the cochlear duct elongation during the development. Given that the curvature and thickness of the medial epithelial layer reflect the point of luminal nuclear stalling, the point is continuously updated in the region at a constant distance away from the growing apex tip of the cochlear duct as described in the text (line: 143) “We found that the curvature and thickness of the MEL changed remarkably more than 100 μm away from the apex tip of the duct along the MEL at any time stage”.

The comment is critical, and thus, we have included the following statements in Discussion section (line 326): “The region of luminal nuclear stalling would reflect the morphological profiles in curvature and thickness of the medial epithelial layer. Therefore, based on our results (Figure 2B), we infer that the region of luminal nuclear stalling is continuously updated at a constant distance away from the growing apex tip of the cochlear duct.”

Another possible inclusion would be a discussion about pitch - nuclear stalling cannot really explain pitch, or there other mechanisms that may account for this?

We understand that the ‘pitch’ in this context means the degree of freedom in the roof-floor axis of the cochlear duct. Given that, our previous work (2021, Ishii et al., eLife) would give clues to answer this. We have previously shown that helical cellular flows occur from the base-floor side to the apex-roof side during the cochlear development. This helical cellular motion gives rise to torsional forces that contribute to the spiral morphogenesis of

the cochlear duct, as mentioned in the latest review by Cohen and Sprinzak (2021, Cohen and Sprinzak, Biophysica Journal). According to your comments, we have included descriptions in Discussion section in the updated manuscript as follows (line 346): “Note that stalling interkinetic nuclear migration would contribute to the bending but not to the torsion of the spiral formation. The torsional force can be generated by the helical cellular flows from the base-floor region to the apex-roof region on the lateral side of the developing cochlear duct [29,47].”

Reviewer: 2

(1) The statement is interesting that the luminal nuclear stalling in the interkinetic nuclear migration is related to morphogenesis of tissues. However, I am wondering how long the luminal nuclear stalling continues. Is it temporary behavior during morphogenesis of developing cochlea? How about is it observed in the cochlea of adult animals? If the authors have any data or suggestive information about this question, they are expected to describe.

We appreciate your comments. We’ve considered that the luminal nuclear stalling would hardly occur in adult animals since we’ve claimed that it is a driver for the morphogenesis. According to the comment, we have included the following statements in Discussion section (line 301): “This dynamic process is observed during the developmental process, but not occur in the adult.”

(2) In morphogenesis of the cochlea, the authors described that, for elongation of the cochlear duct, sonic hedgehog signaling, fibroblast growth factor signaling and non-canonical Wnt–planar cell polarity signaling were required. In addition, they elucidated that the luminal nuclear stalling caused cochlear duct bending. On the other hand, animals have a pair of the cochleae, which are mirror image with each other. What defines chirality of the cochleae of left and right sides of the body. I would like to hear what the authors think about this problem?

To answer this question, a comprehensive understanding of tissue axes determination is required, but it is still an open question (please see Bok et al., Int. J. Dev. Biol (2007) for the details; it is a bit old review but the situation has not changed till date). We refrain from including detailed descriptions in the manuscript as it is not the focus of this study. However, we’ve agreed that your question is a general interest of readers. Accordingly, we have revised the sentence in Introduction section as follows (line 63): “During mouse development, the epithelial cochlear duct initiates from a subregion of the otocyst and extends to form a half turn by embryonic day (E) 12, according to the tissue axes determined in an earlier phase [9,10].”